# The Effects of Solvent on Superhydrophobic Polyurethane Coating Incorporated with Hydrophilic SiO_2_ Nanoparticles as Antifouling Paint

**DOI:** 10.3390/polym15061328

**Published:** 2023-03-07

**Authors:** Ramesh Kanthasamy, Mohammed Algarni, Leo Choe Peng, Nur Ain Zakaria, Mohammed Zwawi

**Affiliations:** 1Department of Chemical and Materials Engineering, Faculty of Engineering Rabigh, King Abdulaziz University, Rabigh 21911, Saudi Arabia; 2Department of Mechanical Engineering, Faculty of Engineering Rabigh, King Abdulaziz University, Rabigh 21911, Saudi Arabia; 3School of Chemical Engineering, Engineering Campus, Universiti Sains Malaysia, Nibong Tebal, Pulau Pinang 14300, Malaysia

**Keywords:** polyurethane, solvent, silica nanoparticles, spray coating

## Abstract

Polyurethane (PU) paint with a hydrophobic surface can be easily fouled. In this study, hydrophilic silica nanoparticles and hydrophobic silane were used to modify the surface hydrophobicity that affects the fouling properties of PU paint. Blending silica nanoparticles followed by silane modification only resulted in a slight change in surface morphology and water contact angle. However, the fouling test using kaolinite slurry containing dye showed discouraging results when perfluorooctyltriethoxy silane was used to modify the PU coating blended with silica. The fouled area of this coating increased to 98.80%, compared to the unmodified PU coating, with a fouled area of 30.42%. Although the PU coating blended with silica nanoparticles did not show a significant change in surface morphology and water contact angle without silane modification, the fouled area was reduced to 3.37%. Surface chemistry could be the significant factor that affects the antifouling properties of PU coating. PU coatings were also coated with silica nanoparticles dispersed in different solvents using the dual-layer coating method. The surface roughness was significantly improved by spray-coated silica nanoparticles on PU coatings. The ethanol solvent increased the surface hydrophilicity significantly, and a water contact angle of 18.04° was attained. Both tetrahydrofuran (THF) and paint thinner allowed the adhesion of silica nanoparticles on PU coatings sufficiently, but the excellent solubility of PU in THF caused the embedment of silica nanoparticles. The surface roughness of the PU coating modified using silica nanoparticles in THF was lower than the PU coating modified using silica nanoparticles in paint thinner. The latter coating not only attained a superhydrophobic surface with a water contact angle of 152.71°, but also achieved an antifouling surface with a fouled area as low as 0.06%.

## 1. Introduction

Superhydrophobic coatings protect clothes, machines, vehicles, buildings, structures and more from extreme weather. They have gained more attention since they exhibit not only non-wetting properties, but also self-cleaning, anti-corrosion, and antibacterial properties. The water contact angle on superhydrophobic coatings is well described by the Cassie–Baxter model [1]. The rough surface of the superhydrophobic coating with low surface energy traps air to minimize the water contact, yielding a water contact angle higher than 150°.

Inorganic nanoparticles are popular in the construction of rough surfaces on superhydrophobic coatings, including in commercial paints. Karmouch and Ross [2] spray-coated a wind turbine with a mixture of commercial silence resin, acrylic polymer, and silica nanoparticles in toluene. After curing at 100 °C for 10 min, the superhydrophobic coating formed due to the low surface energy of epoxy and the roughness created by the silica nanoparticles. However, the surface energy of some polymers may not be sufficiently low for the formation of a superhydrophobic surface. Silica nanoparticles were modified with oligodimethylsiloxane before blending into poly (methyl methacrylate), commercial acrylic paint, or polyfluorene for creating a superhydrophobic coating on different substrates, including glass, paper, and more [3]. The water-based superhydrophobic coating was later developed by incorporating hydrophobic silica nanoparticles and polyvinylpyrrolidone into the ethanol–water solvent. Nevertheless, the functionality and durability of this water-based coating have not been studied. Zhang et al. [4] bound hydrophobic silica on graphene using dopamine and then press-coated the mixture on carbon steel with fluorocarbon paint to form a superhydrophobic surface. Lu et al. [5] later reported on the superhydrophobic titanium oxide (TiO_2_) coating with satisfactory durability. The TiO_2_ nanoparticles with bimodal particle size distribution were dispersed in ethanol that contains perfluorooctyltriethoxysilane before spray coating on adhesive or paint. The water contact angle on the superhydrophobic TiO_2_ coating remained higher than 150° after 40 cycles of the sandpaper abrasion test. Our previous work [6] confirmed that the superhydrophobic rice husk ash coating prepared using a similar method could reduce water uptake, sorption and penetration into the concrete. Hydrophobic silica nanoparticles and copper (I) oxide (Cu_2_O) nanoparticles were also incorporated into the commercial epoxydic paint before coating on top of a thermally sprayed aluminum coating [7]. In addition to the superhydrophobic surface, the modified epoxydic paint showed satisfactory antibacterial properties. The metal–organic framework, zeolitic imidazolate framework-8 (ZIF-8) with a slightly large particle size (0.51 and 0.73 nm), was modified with 1H,1H,2H,2H-perfluorooctyltriethoxysilane before being spray-coated on top of the epoxy-polyamide coating [8]. The superhydrophobic ZIF/epoxy-polyamide coating showed self-cleaning, anti-corrosion, and anti-icing properties that are most desired in extreme weather.

The use of fluorocarbon-based chemicals has been reduced since more concerns about paint sustainability have been raised in recent years. Alumina nanoparticles were functionalized with stearic acid in 2-propanol after being refluxed with toluene [9]. The hydrophobic alumina nanoparticles suspension was then spray-coated on epoxy coating near the curing time. A water contact angle lower than 150° was recorded, indicating the Wenzel state. Cao et al. [10] blended silica nanoparticles into supramolecular silicone polymer synthesized from polydimethylsiloxane, dopamine hydrochloride, and isophoronediisocyanate to form a self-healing coating with a superhydrophobic surface. The superhydrophobic surface was retained after the durability test because of the dynamic intermolecular crosslinking and chemical reorganization in the polymer. Tang et al. [11] coated zinc oxide (ZnO) seeds on PDMS for the growth of ZnO nanorods through a hydrothermal process. The PDMS-ZnO thin film was further doped with Au nanoparticles in a photo-reduction reaction. Without a hydrophobic agent, the PDMS-ZnO/Au thin films still attained a superhydrophobic surface, besides antibacterial properties with photocatalytic and mechanical disinfection effects.

Among the commercial paint and coating materials, polyurethane (PU) coatings have been widely used as a thermal barrier, anti-corrosion, weatherproofing, and an abrasion protection layer. West et al. [12] pre-treated the PU coating using oxygen/argon plasma before fluoroalkyl silane coating. The plasma pretreatment introduced more functional groups on the polymer surface to interact with silane, besides creating hierarchical roughness. Hence, the superhydrophobic PU coating was successfully created. Carreño et al. [13] used N,N-hexamethyldisilazane modified silica nanoparticles to modify PU paint. The superhydrophobic surface only formed when the silica suspension in tetrahydrofuran was spray-coated on top of the PU paint before the gel time was reached. Without sinking into the paint, the hydrophobic silica nanoparticles adhered on top of the paint could generate a highly rough surface with superhydrophobicity. Najafpour et al. [14] blended hexamethyldisilazane-modified silica nanoparticles into acetone containing PU and cetyltrimethylammonium bromide that worked as the surfactant. A hydrophobic PU coating formed after spray-coating, while the superhydrophobic surface was only created when the hydrophobic silica nanoparticles were coated through the electrophoretic deposition. The condensation heat transfer was slightly improved on the superhydrophobic coating due to the extension of droplets sweeping at higher wall subcooling temperature.

Many methodologies have been proposed to develop superhydrophobic coatings, but many of them are difficult to adopt in the production of commercial paints such as PU paints. The main objective of this work is to enhance the surface roughness of commercial PU paint using silica nanoparticles without silane for the creation of a superhydrophobic surface. It is hypothesized that the subsequent spray coat of a solvent on the PU coatings can further enhance the surface roughness by improving the embedment of silica nanoparticles. The effects of unmodified silica nanoparticles and silane-modified silica nanoparticles on the coating hydrophobicity were first compared in the blending-spray coating method. Then, the effects of solvents on the coating hydrophobicity were investigated in the dual-layer spray coating method. Due to the variation of paint solubility in different solvents, the embedment of silica nanoparticles was expected to change in the dual-layer spray coating. Surface roughness and hydrophobicity would be affected. In addition to characterization, the coatings were further studied using the fouling test. The fouled area was computed using ImageJ for quantitative comparison.

## 2. Materials and Methods

### 2.1. Materials

The commercial PU paint, thinner, and hardener were acquired from an undisclosed paint manufacturer. The paint mainly contains 2-methylpropyl ester, 2-butanone, butyl ester, 1-methoxy acetate, 2-propanol, acetic acid, naphtha, and aluminium. The thinner is a mixture of 2,6-dimethyl-4-heptanone, acetic acid ethyl ester, 4-methyl-2-pentanone, and n-hexane, while the hardener is a mixture of acetic acid butyl ester and 1,6-diisocyanate hexane. The silica nanoparticles were supplied by US Research Nanomaterials, Inc. Solvents such as tetrahydrofuran (>99.5%) and ethanol (>99.9%), as well as hydrophobic silane (POTS, 98%), were purchased from Sigma Aldrich.

### 2.2. Blending-Spray Coating Methodology

The components of the coating solution, namely paint, hardener, and thinner, were mixed by the weight ratio of 10:8:8–11 based on the established coating formula provided by the undisclosed industrial collaborator. The paint, hardener, and thinner were mixed using a planetary mixer (Mazerustar KK-250S, Kurabo Electronics, Tokyo, Japan) for 5 min to ensure homogeneity. The solution was spray-coated on glass slides using a spray gun with 1 mm nozzle at a pressure of 2 bar. The distance between the glass slides and the spray gun nozzle was fixed at 20 cm with an angle of 30° from vertical. Then, the coated glass slides were cured in the oven for 5 h at 50 °C. Without silica, the sample is designated as R-0.

Silica nanoparticles (particle size of 20–30 nm) were blended into the coating solution with varied loading (5 wt.% or 7 wt.%) as summarized in Figure 1. The silica nanoparticles were first dispersed in the thinner using an ultrasonicator operated for 30 min. The thinner solution containing silica nanoparticles was then mixed with paint and hardener for 5 min using a planetary mixer. The coating solution was spray coated on the glass slides and cured for 5 h at 50 °C. These samples were labeled as N/Z, where Z represents the silica loading (wt.%).

The effects of silane modification were studied using N/5.0 and N/7.0 samples. The samples were immersed in a mixture of POTS-ethanol at the volume ratio of 1 mL:50 mL for 30 min. After drying for 1 h at room temperature, the silane-modified samples were named S/5.0 and S/7.0, respectively.

### 2.3. Dual-Layer Spray Coating Methodology

In the dual-layer spray coating methodology, the first layer of PU coating was prepared according to Section 2.2. The second layer of coating solution was prepared by dispersing varied loadings of silica nanoparticles (0.1 wt.%, 0.5 wt.%, 1 wt.%, and 2 wt.%) in different types of solvent, namely thinner, ethanol, and tetrahydrofuran (THF), as summarized in Figure 1. Silica nanoparticles with the particle size of 20–30 nm were used. The solution was ultrasonicated for 30 min to achieve homogenous dispersion. The partially cured PU coating (1 h 30 min, 50 °C) was sprayed with a nanosilica-solvent mixture using a spraying bottle. The distance from the spray bottle nozzle to the glass slide was 15 cm at an angle of 45° from vertical. Then, the sample was fully cured for another 3 h and 30 min at 50 °C in the oven. These samples were labeled as X/Y/Z, where X represents the type of solvent (E denotes ethanol, T denotes THF, P denotes paint thinner, Y represents the particle size (20 nm), and Z is silica loading (wt.%).

### 2.4. Coating Characterization

The spectra from 600 cm^−1^ to 3800 cm^−1^ were analyzed using attenuated total reflection-Fourier transformed infrared (FT-IR) spectroscopy (Nicolet iS10, Thermo Scientific, Waltham, MA, USA) to study the chemical components in the coating samples. The spectra were obtained from 32 scans at a resolution of 4 cm^−1^ using a diamond crystal placed on the dried and ground coating. The surface morphology of the coating samples was analyzed via a scanning electron microscope (SEM) (HITACHI TM3000, Hitachi Ltd., Minato-ku, Tokyo, Japan) with a high-sensitive semiconductor BSE detector. The coating surface was sputter-coated with a thin conductive gold layer to improve the stability of the chargeable sample. Superficial roughness was studied with an atomic force microscope (AFM, Bruker Multimode 8, Bruker, Billerica, MA, USA) with a scan size of 10 µm × 10 µm under tapping mode. The analysis software used was Nanoscope Analysis 1.7. Contact angle measurement was conducted using the image of a water droplet (10 µL) placed on the coating surface at ambient conditions. The electronic microscope (1000X Electronic Digital Microscope Handheld USB Magnifier) was used to capture the image of the water droplet on the coating surface. The water contact angle of the water droplet was then measured using ImageJ software. The average water contact angle was determined from 3 replicates of measurement on each sample.

### 2.5. Fouling Test

The antifouling property of the coatings was investigated by dipping the coating samples in the slurry solution. The slurry solution was prepared by mixing 35 g of kaolinite and 0.1 g of methylene blue dye in 100 mL of distilled water. The mixture was stirred for 15 min and ultrasonicated for 30 min to obtain well-dispersed kaolinite and dye in distilled water. Then, the glass slides with coating samples were dipped in the mixture and dried at 50 °C for 20 h. The fouled area was then determined using ImageJ software.

## 3. Results and Discussion

### 3.1. Characteristics of PU Coatings Incorporated with Silica Nanoparticles

The FTIR spectra of PU coating (R-0), PU coating blended with silica nanoparticles (N/7.0), and PU coating blended with silica nanoparticles and modified with silane (S/7.0) are shown in Figure 2. Meanwhile, Figure 3 shows the dual-layer PU coatings with 2.0 wt.% of silica in different solvents. The N/7.0 sample exhibited significant peaks of the hydroxyl stretching of the Si-OH (3384.60 cm^−1^) and water deformation band (1686.52 cm^−1^) after blending the hydrophilic nanoparticles into PU coating [15]. These peaks disappeared after silane modification, as shown by the S/7.0 sample. The small peak at 1114 cm^−1^ of the S/7.0 sample further indicates the Si-C-O and -CF bonds of POTS [14,15]. The S/7.0 sample had been successfully modified using POTS. The E/20/2.0 and T/20/2.0 samples did not show any peak at 3382.67 cm^−1^ due to the absence of hydroxyl stretching of Si-OH [15]. The silica nanoparticles were either not successfully bound with or embedded too deep into the polymer matrix. The P/20/0.2 sample showed significant peaks of the hydroxyl stretching of the Si-OH (3382 cm^−1^) and water deformation band (1685 cm^−1^) [16]. The FTIR spectrum of the P/20/0.2 sample prepared using the dual-layer coating method had a similar pattern compared to N/7.0 sample, which was prepared by blending silica nanoparticles into a PU coating.

Moreover, the C-O absorption (1237 cm^−1^ and 1243 cm^−1^), C-N absorption (1378 cm^−1^ and 1381 cm^−1^), and N-H absorption (1461 cm^−1^, 1463 cm^−1,^ and 3384 cm^−1^) peaks existed in all coating samples because of the use of PU. The peak at 2932 cm^−1^ could be attributed to the C-H stretching vibration [17,18,19]. The wide signal around 1686 cm^−1^ shows the isocyanurate group obtained from the hardener solution, while the 1727 cm^−1^ peak indicates the urethane carbonyl group obtained from the PU coating [13].

Figure 4 shows the SEM images of the unmodified and modified PU coatings. Figure 4a displays a smooth surface of the unmodified PU coating surface as silica nanoparticles were not blended or coated on its surface. The surface only exhibited polymer aggregates. The incompatibility between the hydrophobic PU and the hydrophilic silica nanoparticles caused the formation of silica agglomerates. As shown in Figure 4b, the silica agglomerates appeared on the PU coating surface after blending. The silica agglomerates were covered by the silane in the post-modification of the PU coating blended with silica nanoparticles (Figure 3c). Silane modification involves the hydrolysis and condensation of silanol groups. The condensed silanol groups covered the coating surface, as reported in our previous works [20,21,22,23]. Figure 4d–f show the dual-layer coatings with PU and 2 wt.% of silica nanoparticles in the different solvents. All these coatings showed the adhesion of silica nanoparticles. The amount of nanoparticles adhered to the PU coatings increased from the E/20/2.0 sample to the P/20/2.0 sample, followed by the T/20/2.0 sample. Ethanol only caused PU to swell, but the paint thinner was formulated to dissolve and reduce the viscosity of PU used in this study effectively. Compared to other solvents, such as acetone, cyclohexane, toluene, benzene, and methyl ethyl ketone, THF was considered a very strong solvent for polyurethane [24]. Monaghan and Pethrick [25] commented that the curing of PU could be significantly affected by the solvents. Hence, the adhesion of silica nanoparticles on the PU coating before complete curing was affected by the solvent selection in this study. The surface morphology of the coatings prepared using different solvents was further studied using AFM. Figure 5a shows the surface morphology of the PU coating without modification. It attained a low root mean squared roughness (*R_q_*) value of 4.15 nm.

The surface is considerably smoother due to the absence of silica nanoparticles. As shown in Figure 4b–d, the topography of the modified PU coatings changed remarkably after coating with the silica nanoparticles. The Rq value increased to 13 nm when silica nanoparticles dispersed in ethanol were coated. The silica nanoparticles were homogenously coated on the E/1.0 sample, as shown in the SEM image (Figure 4d), creating excellent roughness. Although more silica nanoparticles adhered to the modified PU coatings using THF (Figure 4e), the surface roughness only increased to 6.11 nm. THF is a strong solvent that can dissolve PU easily, and the silica nanoparticles were embedded into the PU coating.

Paint thinner is still the best solvent to coat silica nanoparticles on PU coating. The highest surface roughness was recorded with an *R_q_* value of 15.0 nm. Paint thinner interacted with PU satisfactorily, creating additional roughness that could be observed from the topography of the P/1.0 sample shown in Figure 5d.

PU coating is hydrophobic in nature. Hence, a water contact angle higher than 90° on the unmodified PU coating (R-0) was recorded. The water contact angle reduced as the loading of silica nanoparticles in the R-0, N/5.0, and N/7.0 samples was raised from 0 to 7.0 wt.% in the blending method, as shown in Table 1. With silane modification, the water contact angle on S/5.0 and S/7.0 samples attained a higher water contact angle than the unmodified PU coating (R-0). Similar to the study by Jiang et al. [26], the water contact angle increased when the content of the silane–silica nanoparticles was increased. Nevertheless, these samples only exhibited a hydrophobic surface as their water contact angle fell between 90° and 150°. Although silane was used to reduce the surface energy of the PU coating, blended with silica nanoparticles, the roughness creation was still insufficient in these samples for achieving superhydrophobicity [27].

Blending caused the embedment of silica nanoparticles into the polymer matrix, as shown in the SEM image (Figure 4b). By changing the coating method from blending-spraying to dual-layer spraying the embedment of silica nanoparticles was successfully reduced, as shown in the SEM images (Figure 4d–f). Hence, the water contact angle on the dual-layer coating with PU and silica nanoparticles in THF or thinner increased significantly (Table 2). The P/20/1.0 and P/20/2.0 samples even attained superhydrophobic surfaces with water contact angles higher than 150°.

As mentioned previously, PU is only soluble in cyclic ether, such as THF and dioxane, or cyclic ketone, such as cyclohexane and cyclohexanone. The paint thinner contains 2,6-dimethyl-4-heptanone, acetic acid ethyl ester, 4-methyl-2-pentanone, and n-hexane, which can dissolve PU more effectively compared to THF. As the silica nanoparticles were sprayed on the first coating layer and fully cured, they effectively adhered to the polymer matrix and created roughness for the superhydrophobic surface [13]. Contrariwise, Zheng et al. [28] pre-treated silica nanoparticles using hydrophobic heptadecafluorodecyltriethoxysilane before spray coating on the waterborne PU coating to achieve a water contact angle up to 172.2 ± 1.6°. However, the water contact angle on the dual-layer coatings with PU and silica nanoparticles in ethanol (E/20/0.1, E/20/0.5, E/20/1.0, and E/20/2.0 samples) was reduced significantly by increasing the silica loading in ethanol. Ethanol is not an effective solvent for PU. Even worse, ethanol with high polarity could attract moisture and form strong hydroxyl bonds, resulting in a hydrophilic surface. Comparing two coating methods used in this study (Table 3), it is clear that only dual-layer coatings with PU and silica nanoparticles in paint thinner (P/20/2.0) showed small variability. The roughness created on this surface was considerably more homogenous than other coatings, as the silica nanoparticles could have adhered well on PU when the appropriate solvent to interact with PU was used.

### 3.2. Antifouling Properties of PU Coatings

The antifouling properties of the coating surfaces were further studied using the kaolinite slurry solution blended with dye as the simulated contaminant. The image was analyzed using ImageJ so that the fouling could be quantified and compared precisely. Figure 6a–d shows the fouling areas of the PU coating (R-0), PU coating blended with silica nanoparticles (N/7.0), silane-modified PU coating blended silica nanoparticle (S/7.0), and dual-layer coating with PU and silica nanoparticles in paint thinner (T/1.0) before and after the fouling test.

Based on Figure 6c, the slurry adhered severely to the PU coating blended with silica nanoparticles (S/7.0) even after silane modification, although the surface hydrophobicity was slightly increased. The fouled area accounts for up to 98.80% of the total area. The PU coating blended with silica nanoparticles (N/7.0) showed remarkable improvement in antifouling properties compared to the unmodified PU coating (R-0). The fouled area was reduced from 30.42% to 3.37%, even though the water contact angle was not significantly reduced. The chemical properties of the coating surface could be the significant factor that affected its fouling properties. Silane modification should be adequately studied to minimize fouling. The PU coating coated with silica nanoparticles in paint thinner using the dual-layer coating method (T/1.0) attained the lowest fouled area, only 0.06%. This superhydrophobic surface coating without silane modification remained clean, and almost no slurry stain adhered to the surface, as shown in Figure 6d.

## 4. Conclusions

Two modification methods were compared in the development of antifouling PU coating, namely blending and dual-layered coating methods. The PU coating blended with silica nanoparticles (20–30 nm) showed significant surface chemistry changes. The FTIR spectrum showed the existence of hydroxyl stretching of the Si-OH and water deformation band. The silica nanoparticles were embedded into the PU coating, as shown in the SEM image, resulting in the insignificant change of the water contact angle. However, the coating could prevent fouling by kaolinite slurry. The fouled area was greatly reduced to 3.7% after blending silica nanoparticles. The fouled area of the unmodified PU coating was estimated to be 30.42% after the fouling test. After silane modification, the PU coating blended with silica nanoparticles was severely fouled. The fouled area was increased to 98.80% after modification using POTS. The water contact angle on the PU coating was slightly increased (10.60%) after incorporating silica nanoparticles and POTS modification. The surface chemistry changes affected the fouling properties significantly. Using the dual-layer coating method, silica nanoparticles in different solvents were coated on PU coatings. The silica nanoparticles were homogeneously coated on PU coating when ethanol was used as the solvent, and the surface roughness was significantly enhanced (222.89%). However, the water contact angle was significantly reduced to 18.04° due to the introduction of hydroxyl groups. The superhydrophobic PU coating only formed when it was coated with silica nanoparticles in paint thinner. The surface roughness was increased to 15.0 nm using paint thinner, but only 6.11 nm using THF with strong dissolution properties. The silica nanoparticles were embedded in the PU paint when THF was used as the solvent in the dual-layer coating method. Hence, paint thinner was the preferred solvent of silica nanoparticles to produce a superhydrophobic PU coating (P/20/1.0) through the dual-layer coating method.

## Figures and Tables

**Figure 1 polymers-15-01328-f001:**
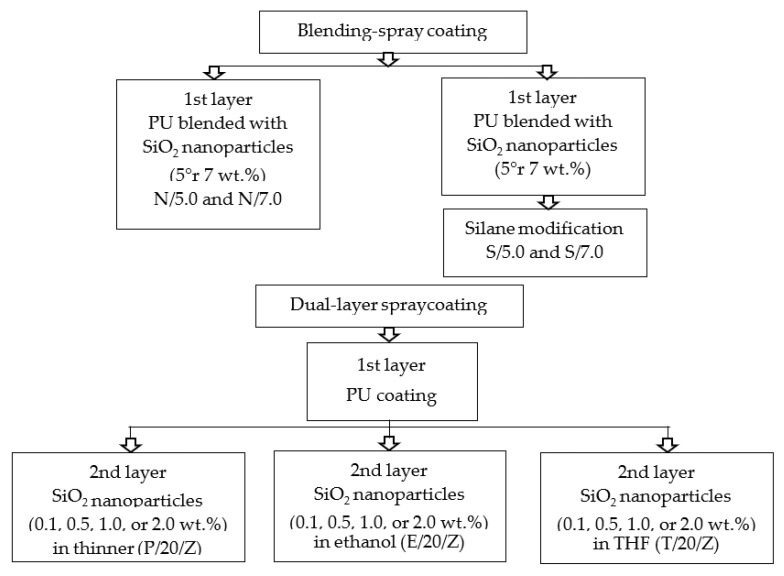
Summary of blending-spray coating and dual-layer spray coating methods.

**Figure 2 polymers-15-01328-f002:**
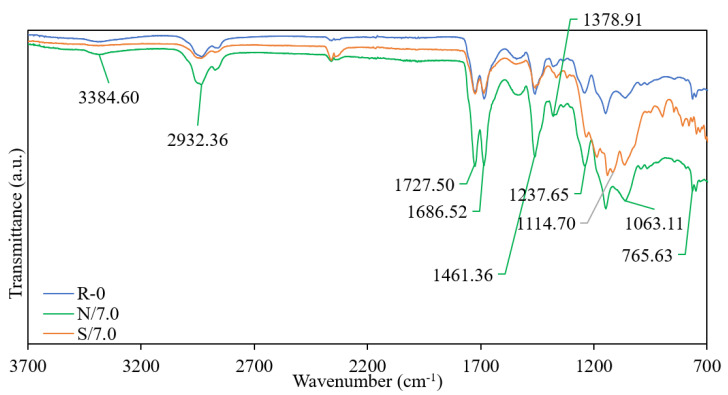
FTIR spectra of PU coating (R-0), PU coating blended with silica nanoparticles (N/7.0), and PU coating blended with silica nanoparticles and modified with silane (S/7.0).

**Figure 3 polymers-15-01328-f003:**
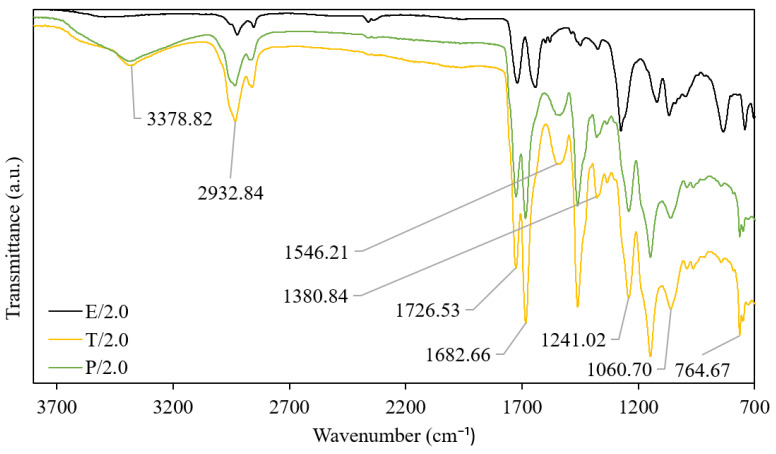
FTIR spectra of dual-layer PU coating with 1.0 wt.% of silica nanoparticle in different solvents (ethanol—E/1.0, THF—T/1.0, paint thinner—P/1.0).

**Figure 4 polymers-15-01328-f004:**
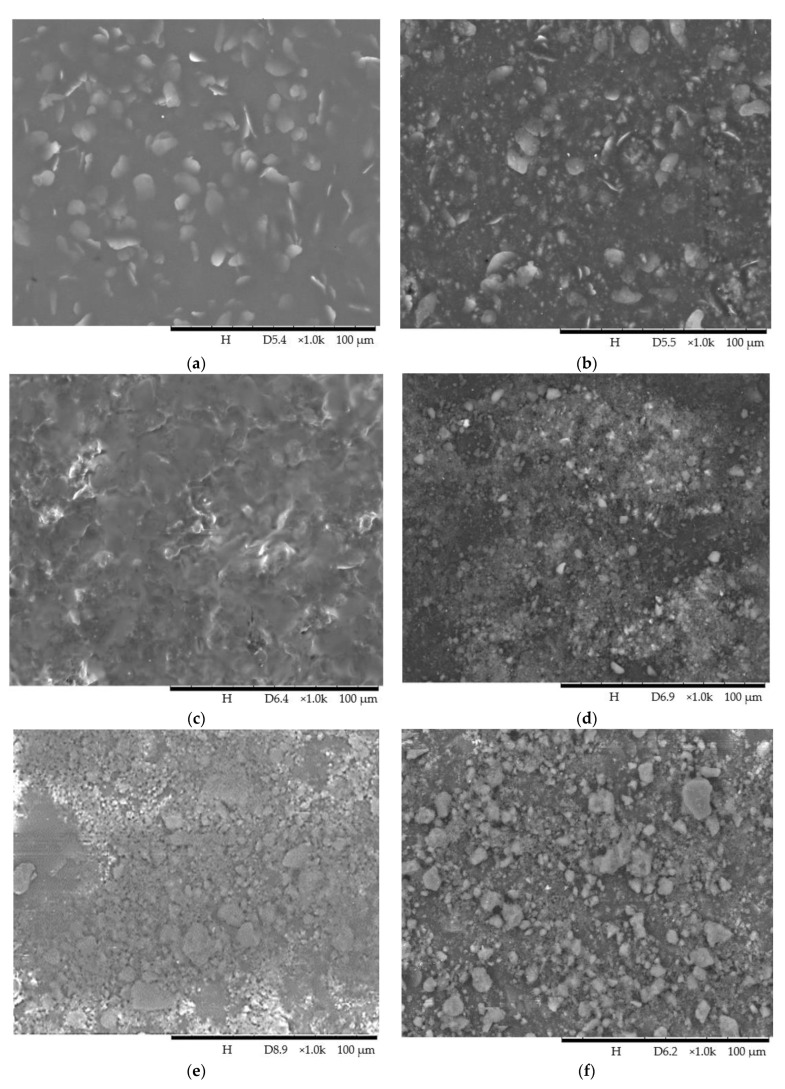
SEM images of (**a**) PU coating (R-0), (**b**) PU coating blended with silica nanoparticles (N/7.0), (**c**) silane modified PU coating blended with silica nanoparticles (S/7.0), dual-layer coating with PU and 1.0 wt.% of silica nanoparticles in (**d**) ethanol (E/1.0), (**e**) THF (T/1.0) and (**f**) paint thinner (P/1.0).

**Figure 5 polymers-15-01328-f005:**
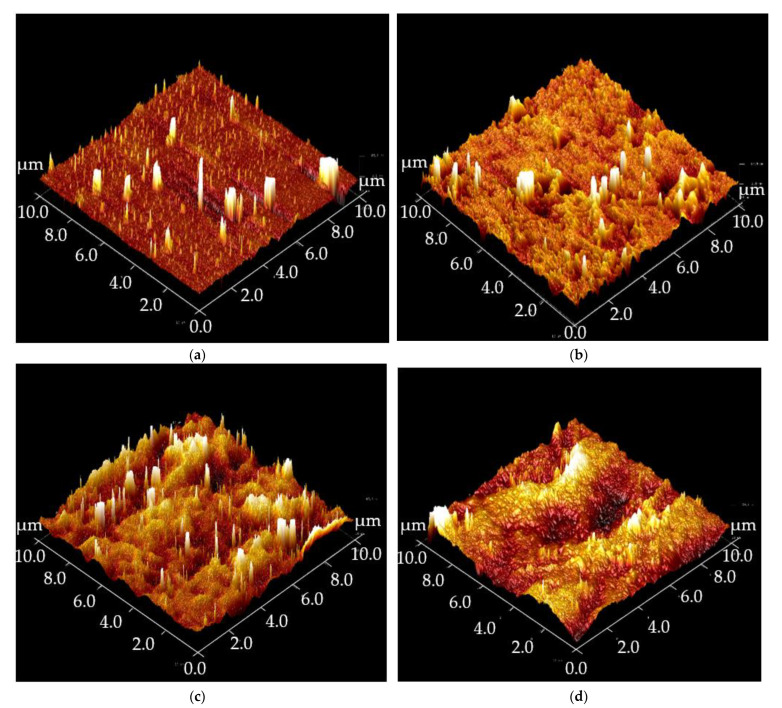
The average roughness in AFM images of (**a**) PU coating without modification (R-0: 4.15 nm), and (**b**) PU coatings modified using silica in ethanol (E/1.0: 13.4 nm), (**c**) THF (T/1.0: 6.11 nm), or (**d**) paint thinner (P/1.0: 15 nm).

**Figure 6 polymers-15-01328-f006:**
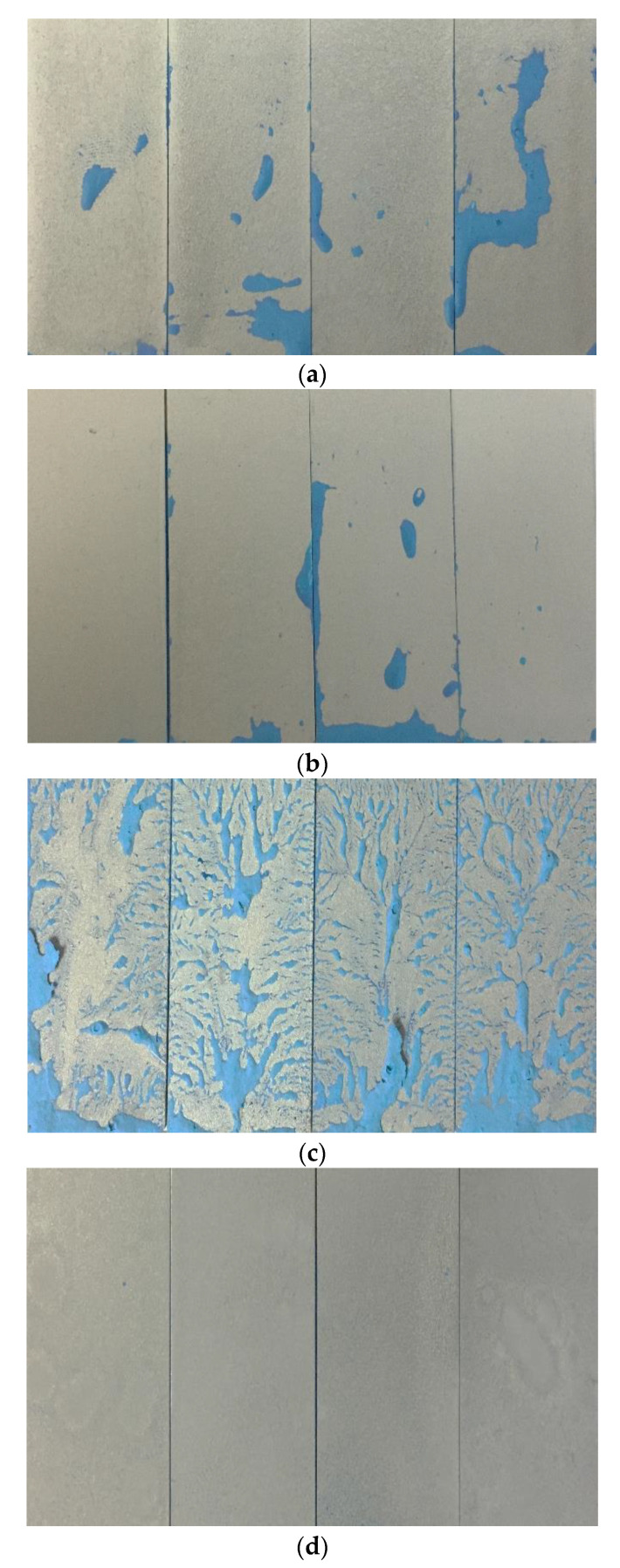
Fouled area of (**a**) PU coating (R-0: 30.42%), (**b**) PU coating blended with silica nanoparticle (N/7.0: 3.37%), (**c**) silane-modified PU coating blended silica nanoparticle (S/7.0: 98.80%) (**d**) and dual-layer coating with PU and silica nanoparticles in paint thinner (T/20/1.0: 0.06%).

**Table 1 polymers-15-01328-t001:** WCA of PU coatings with varied loadings of silica nanoparticles prepared using the blending-spraying coating method.

Sample	Silica Loading (wt.%)	WCA (°)
R-0	0	99.62 ± 3.38
N/5.0	5.0	98.54 ± 10.85
N/7.0	7.0	94.72 ± 6.00
S/5.0	5.0	104.03 ± 14.40
S/7.0	7.0	110.17 ± 4.85

**Table 2 polymers-15-01328-t002:** WCA of PU coating with varied loadings of silica nanoparticles in ethanol, THF, or thinner prepared using dual-layer coating using different types of solvent for the second layer of coating.

Sample	Silica Loading (wt.%)	Solvent	WCA (°)
E/20/0.1	0.1	Ethanol	69.81 ± 10.68
E/20/0.5	0.5	Ethanol	53.94 ± 6.34
E/20/1.0	1.0	Ethanol	37.74 ± 19.92
E/20/2.0	2.0	Ethanol	18.04 ± 1.83
T/20/0.1	0.1	THF	100.69 ± 3.03
T/20/0.5	0.5	THF	123.86 ± 13.12
T/20/1.0	1.0	THF	143.95 ± 10.19
T/20/2.0	2.0	THF	148.81 ± 2.96
P/20/0.1	0.1	Thinner	85.87 ± 5.41
P/20/0.5	0.5	Thinner	142.00 ± 2.79
P/20/1.0	1.0	Thinner	153.08 ± 0.51
P/20/2.0	2.0	Thinner	152.71 ± 1.81

**Table 3 polymers-15-01328-t003:** Average WCA of PU coating prepared using different methods.

Samples	Method	Chemicals	Average WCA (°)
R-0	Spray coating	-	99.62 ± 3.38
N/7.0	Blending-spray coating	-	94.72 ± 6.00
S/7.0	Blending-spray-silane post-treatment	perfluorooctyltriethoxy silane	110.17 ± 4.85
E/20/2.0	Dual-layer spray coating	ethanol	18.04 ± 1.83
T/20/2.0		tetrahydrofuran	148.81 ± 2.96
P/20/2.0		paint thinner	152.71 ± 1.81

## Data Availability

Not applicable.

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
