# Peer review of "The Effects of Solvent on Superhydrophobic Polyurethane Coating Incorporated with Hydrophilic SiO2 Nanoparticles as Antifouling Paint"

_polymers, 2023, doi:10.3390/polym15061328_

Round 1
Reviewer 1 Report
There are lots of typo, lack of citations in this article. The structure of introduction part is not attractive and lack of research motivation and hypothesis.
1. Typos everywhere. e.g., Line 132-133 degree C symbol and degree symbol. Line 160 cm-1....
2. Don't use brand name or unfamiliar name without interpretation. e.g., kaolin is not an appropriate scientific term, if you still want to use that, please briefly explain that.
3. Characterization section provide too little detail of instrumentation. e.g., is FTIR using ATR or other mode? How did you prepare FTIR samples? which detector was used in SEM? BSE or SE? which mode was used in AFM? contact? tapping? what is the model and the brand of electronic microscope? How did you prepared the SEM samples (freeze fracture? microtome? or smooth surface)? and any sputter coating?
4. Lack of flow chart illustrating entire research procedures. i.e., box diagram briefly pictorial illustrating your work, to help the reader understand your work easily.
5. Some citation were missing. e.g. Line 40 lack of citation of Cassie-Baxter model
Reviewer 2 Report
The manuscript "polymers-2154351" by Kanthasamy et al. reported the The Effects of Solvent on the Superhydrophobic Polyurethane Coating Incorporated with Hydrophilic SiO2 Nanoparticles as the Antifouling Paint. After review, this study is interesting but the authors have to make minor changes. The authors should refer to the following comments to improve their work:
General comments:
a. The language of the manuscript should be checked. There are many errors.
b. According to the results, which do you consider to be the optimal sample? Please report in conclusion.
Specific comments:
Introduction
a. Terms such as TiO2, Cu2O, ZIF-8, ZnO, and PDMS are abbreviated in the introduction without explanation, please revise and rewrite.
b. Page 2, line 64: Cu2O: 2 must be subscript.
c. Page 2, line 58 and 61: TiO2: 2 must be subscript.
Materials and Methods
a. Please be careful in writing units. Page 3, line: 132, 133, and 140. Page 4, line: 153, 155, 156, 160, and 178.
Results and discussion
a. Page 4, line 197: There is no dot at the end of the sentence.
b. Figure 1 and 2: Delete markers or show them as dotted lines.
c. Page 6, line 214: (Fig. 3(c)).
d. Figure 3b: How are nanoparticles visible from a distance of 100μm?
e. Figure 3: What is the relationship between the use of silane and the disappearance of polymer aggregates? In Figure 3c, polymer aggregates are not seen.
Reviewer 3 Report
This paper reports the effects of solvent on the superhydrophobic polyurethane coating incorporated with hydrophilic SiO2 nanoparticles as the antifouling paint. The description of the article is not clear enough and there are many writing and format mistakes.
1. The description of the article is not clear enough. As described in Line 145 that S/0.7 was were prepared from N/0.7, but N/0.7 was not mentioned in the preparation method. S/0.7 was not mentioned and studied in the following.
2. As the PU is hydrophobic and silica nanoparticles is hydrophilic, how can silica nanoparticles disperse uniformly in the PU matrix and how about the stability?
3. Scale bar is needed in Figure 5.
4. Some circles in Figure 1 are not the right places.
5. There are many writing and format mistakes, such as the temperature symbol in Line 133 and 140, suffix in Line 58, 61, 64, and reference in Line 206 [13], [17], [18].
Round 2
Reviewer 1 Report
The authors generally replied the questions in previous review and the quality has improved. Overall is easier for the readers to follow and repeat critical scientific work.
Author Response
The authors would like to thank the reviewer for providing useful comments to improve this article.
Reviewer 3 Report
It is mentioned in last report that "As described in Line 145 that S/0.7 was were prepared from N/0.7, but N/0.7 was not mentioned in the preparation method. S/0.7 was not mentioned and studied in the following." Although Figure 1 is added in the revised version, the problem is still the same. They should use other samples (such as N/7.0 and S/7.0) that used in the coating as an example.
